# Recycling of Waste Bamboo Biomass and Papermaking Waste Liquid to Synthesize Sodium Lignosulfonate/Chitosan Glue-Free Biocomposite

**DOI:** 10.3390/molecules28166058

**Published:** 2023-08-15

**Authors:** Qingzhi Ma, Guiyang Zheng, Jinxuan Jiang, Wei Fan, Shengbo Ge

**Affiliations:** 1The Archives, Henan Agricultural University, Zhengzhou 450002, China; 2Co-Innovation Center of Efficient Processing and Utilization of Forest Resources, College of Materials Science and Engineering, Nanjing Forestry University, Nanjing 210037, China; nanlinguiyangzheng@njfu.edu.cn (G.Z.); jxjiang@njfu.edu.cn (J.J.); 3School of Textile Science and Engineering & Key Laboratory of Functional Textile Material and Product of Ministry of Education, Xi’an Polytechnic University, Xi’an 710048, China; fanwei@xpu.edu.cn; 4Aerospace Kaitian Environmental Technology Co., Ltd., Changsha 410100, China

**Keywords:** waste reuse, biocomposite, high performance, sodium lignosulfonate

## Abstract

The development of the paper industry has led to the discharge of a large amount of papermaking waste liquid containing lignosulfonate. These lignin black liquids cause a lot of pollution in nature, which runs counter to the current environmental protection strategy under the global goal. Through the development and use of lignosulfonate in papermaking waste liquid to increase the utilization of harmful substances in waste liquid, we aim to promote waste liquid treatment and reduce environmental pollution. This paper proposes a new strategy to synthesize novel glue-free biocomposites with high-performance interfacial compatibility from papermaking by-product sodium lignosulfonate/chitosan (L/C) and waste bamboo. This L/C bamboo biocomposite material has good mechanical properties and durability, low formaldehyde emissions, a high recovery rate, meets the requirements of wood-based panels, and reduces environmental pollution. This method is low in cost, has the potential for large-scale production, and can effectively reduce the environmental pollution of the paper industry, promoting the recycling of biomass and helping the future manufacture of glue-free panels, which can be widely used in the preparation of bookcase, furniture, floor and so on.

## 1. Introduction

With the massive demand for cultural communication means such as books, manuscripts, and notes, paper is mass-produced as a raw material [1,2]. However, the bleaching process of plant fibers in the paper industry results in a large amount of wastewater containing lignosulfonates [1,3]. The direct discharge of this black liquor lignin causes great pollution by organic and toxic substances into aquatic environments (Figure 1) [4,5]. Papermaking black liquor is usually recovered by cooking and extracting solids or other substances that promote the decomposition of organic matter, which are added to degrade it into small molecular substances that can be decomposed naturally to meet harmless discharge standards [6,7]. However, the degradation and purification of black liquor reduce the yield of by-products, and the solids extracted by cooking are usually burned due to their high lignin content, which contains a large amount of lignin sulfate and cannot be used, which is not in line with the high-value utilization of waste in current societal demand [8,9].

Lignosulfonates are commonly used in construction as dispersants, although at very low levels [7,10,11]. This method of using as an additive cannot be used on a large scale, so there is no effective utilization of the by-products made from lignin black liquor. As a product of lignin sulfonation, sodium lignosulfonate inherits a three-dimensional structure and diversified functional groups of lignin, and due to the introduction of sulfonic acid groups, sodium lignosulfonate has stronger compatibility than lignin and can be used more easily as a substrate for biomass composites [12,13]. Therefore, the strategy of using sodium lignosulfonate to prepare biomass composites is feasible [14,15,16].

Fibers from coniferous or hardwood wood are used to make paper, contributing to global warming and soil erosion and requiring the development of alternative plant fibers. While bamboo is one of many biomass resources, a straight texture and simple structure make bamboo suitable for composite material production due to its physicochemical properties and short growth cycle [17,18,19,20,21,22]. Therefore, bamboo paper production has been gradually applied in recent years, but the screening process of papermaking raw materials produces a large amount of waste, usually in small branches or peeled shavings, and the usual incineration treatment method obviously does not meet the requirements of green recycling. Therefore, the waste bamboo produced by bamboo papermaking still needs to be recycled [23,24,25,26].

The development of bamboo fiberboard or particleboard might be one of the strategies to recycle waste bamboo from papermaking [27,28]. The common strategy of using adhesives as substrates to prepare bamboo biocomposites generates a lot of air pollution, water pollution, and other pollution problems and also leads to the waste of petroleum resources and difficulty in recycling [29,30,31]. Using a glue-free strategy to prepare waste bamboo composites may be a solution [32]. A glue-free strategy could greatly improve recyclability, and the recycled waste bamboo fibers could be reused to reduce the waste of biomass resources [33,34]. However, the performance of the non-adhesive bamboo fiberboard made by hot pressing is relatively poor at present and cannot adapt to the current high-performance, multi-functional use direction [35,36,37]. Therefore, a suitable modification scheme is needed to improve the compatibility of the two and improve the performance of the material [38,39,40].

Chitosan (CS), as a biomass resource extracted and processed from marine arthropods, has also attracted attention due to its good compatibility [41]. Due to its excellent biological function and good biocompatibility, and hydrophilicity, chitosan can effectively improve the binding of fibers [42]. Here, we used the NaOH solution to pretreat natural bamboo powder by removing hemicellulose and enabling the formation of a tighter combination of lignin and cellulose during hot pressing. Chitosan and sodium lignosulfonate are added to the pretreated bamboo powder by blending, and sodium lignosulfonate and chitosan form a composite structure to connect the fibers under a high temperature and high-pressure environment of hot molding. This improves the mechanical properties and durability of the glue-free bamboo fiberboard (Figure 1).

## 2. Results and Discussion

### 2.1. Mechanical Properties of the L/C Bamboo Biocomposite

Figure 2a shows the tensile strength of different samples. It is not difficult to see from the comparison that the tensile strength of the bamboo biocomposite prepared by alkali pretreatment was significantly improved due to the removal of hemicellulose in bamboo by alkali treatment. This made it easier for the subsequent hot-pressed lignin to deposit on the surface of the fiber to form a tight bond, which enabled the biocomposite made from the pretreated bamboo powder to achieve greater density and better mechanical properties (Figure 2f) [43]. This was not only due to the removal of hemicellulose or the increase in tensile strength (17.13 MPa) and flexural strength (37.07 MPa) (Figure 2b,d) but also due to the pretreatment on the removal, which increased the modulus of elasticity of the material [44,45,46].

The addition of L/C blends improved the tensile strength, and the mechanical properties of the best sample of 15% L/C reached 18.71 MPa, which showed that the addition of blends improved the mechanical properties (Figure 2a,b). Sodium lignosulfonate and chitosan formed a new combination with the pretreated bamboo powder in hot pressing; this novel combination improved the interfacial compatibility of lignin and cellulose, thus enabling the formation of a tighter structure and endowing the material with better mechanical properties [47] (Figure 2f). The characterization of the bending strength also supports this view (Figure 2d,e). After pretreatment and compared with BC (29.55 MPa), AKBC (32.38 MPa) had a smaller increase in flexural strength than tensile strength, while 5% L/C (42.45 MPa), 10% L/C (41.74 MPa), and a mechanical strength of 15% L/C (48.06 MPa) which was significantly improved, showing that the addition of L/C blends could effectively improve the bending strength of the material [48].

By improving the interfacial compatibility between lignin and cellulose in the bamboo powder, it provided sufficient support for lignin as the base material of the material, and the effective combination of the base material, and the reinforcing material was reflected in the improvement of the bending performance [49]. Not only this but compared with BC (4.83 GPa) and AKBC (4.67 GPa) samples without L/C blends, the flexural strength of samples added with L/C blends was more obvious, and 15% L/C reached 5.79 GPa. Figure 2c shows the spectra of different samples under the XRD test and the crystallinity of different samples was calculated according to the spectrum (Table 1); the calculation of the crystallinity is as follows [50]:$$\mathrm{CrI}=\left(1\;-\frac{I_{\mathrm{am}}}{I_{002}}\right)\;\times \;100\%\;$$

I_am_ is the diffraction intensity representing the amorphous background region between the two crystalline peaks; I_002_ is the maximum intensity at the (002) lattice diffraction angle.

It is not difficult to see that after pretreatment, the crystallinity of the sample was significantly improved due to the removal of hemicellulose and cellulose amorphous regions in the alkali treatment, which effectively increased the proportion of cellulose crystallization regions [51,52]. The increase in the crystalline region indicates that the proportion of orderly arrangement inside the fiber increased, which could provide better mechanical strength than the disordered amorphous region [53].

However, the addition of L/C blends decreased the crystallinity of the material because the addition of non-crystalline substances diluted the proportion of cellulose crystalline regions. Therefore, when the amount of L/C added was small, the contribution to fiber bonding was less than the influence of its own structure on crystallinity. Therefore, the crystallinity decreased, but when the L/C addition increased, the compatibility between the fibers and lignin was effectively improved, thus increasing the crystallinity of the material. This could also be corroborated by the characterization of other mechanical properties. Therefore, the addition of a suitable L/C blend can provide a way of utilizing the by-products of papermaking waste liquor [54].

### 2.2. Water Resistance of the L/C Bamboo Biocomposite

The water resistance of biomass composites is one of the conditions where they can be used outdoors [55,56]. Figure 3a shows the contact angles of the prepared samples. It is not difficult to see by comparison that although the contact angle of AKBC prepared by pretreatment was smaller than that of BC at 0 s, it remained stable for a long time until it surpassed BC after 30 s. This was due to the removal of hemicellulose and some lignin after the initial alkali treatment, which exposed the surface hydroxyl groups, and these hydroxyl groups quickly combined with water, resulting in a larger contact angle than that of the untreated BC sample. Additionally, because most of the hydrophilic hemicellulose and amorphous regions of cellulose were removed, and the lignin and cellulose that were separated from the original structure formed a tighter combination, and because most of the hydrophilic hemicellulose and amorphous regions of cellulose were removed, the lignin and cellulose that were separated from the original structure formed a tighter combination [57,58]. Therefore, it can be stated that the alkali treatment was effective [59]. However, it is worth pointing out that through comparison, it was found that the addition of the blend L/C made the value of the contact angle decrease. This was because the L/C blend was still hydrophilic, which caused a certain decrease in the contact angle. This can be seen by comparing it with natural bamboo, but the prepared biomass composite still had a certain hydrophobicity (Figure 3d).

Judging from the thickness and mass change rate of the sample in water, the pretreated sample could effectively resist the water environment, which is the same as the conclusion obtained from the analysis of the contact angle data (Figure 3b,c). However, different from the results of the contact angle, except for 15% L/C, the samples added with L/C blends had good water resistance; that is, they experienced dimensional stability in a water environment, although they all had a high rate of mass change. This was due to the addition of hydrophilic substances to increase the mass increase rate and water contact angle of the material. However, L/C blends can promote the combination of fiber and fiber and fiber and lignin. This boost allowed the material to form a tighter structure. Tise dense structure and close combination enhanced the dimensional stability of the material. Therefore, although the mass increase rate of 5% L/C reached 10%, it still maintained a relatively low dimensional change rate of 5%. Therefore, it can be said that the material had a certain water durability.

### 2.3. Heat Resistance Analysis of the L/C Bamboo Biocomposite

Sufficient thermal stability is a prerequisite for the safe use of materials, especially flammable biomass materials. Appropriate flame retardancy can effectively avoid fires and other disasters that endanger humans and the environment [60]. The thermal conductivity of the material was measured by means of heat flow conduction. The higher the thermal conductivity, the more uniform the temperature distribution, and the safer it was in an outdoor dry and high-temperature environment (Figure 4b). As can be seen in Figure 4a, the pretreated AKBC (0.599 W/mK) was much higher than BC without any treatment (0.386 W/mK). After pretreatment, due to the removal of hemicellulose and cellulose amorphous regions, the bamboo cell wall was deconstructed, and lignin precipitated on the fiber surface to form a more stable filling in the subsequent hot pressing, which gave the material a denser structure. Thermal conductivity was, thus, improved. The thermal conductivity of the sample added with the L/C blend was further improved also due to the improved compatibility of the blend to cellulose and lignin, which enabled the material to be further densified. Among them, 10% L/C could reach 0.648 W/mK, which is a very large improvement compared with BC. This is because thermal conductivity was limited with the addition of the L/C blend. If an excessive amount of the blend was added, the excess L/C would not be able to promote interfacial compatibility, and the degree of thermal conductivity would be reversed.

The sample was placed under the flame of an alcohol lamp to observe its combustion. Similarly, the state of the sample under the flame of the alcohol lamp could also give back the same result (Figure 4c). As a comparison with natural bamboo, it was not difficult to see that there was already an obvious flame at 3 s, and it could burn independently from the flame of the alcohol lamp at 6 s and 9 s, which showed that natural biomass had poor flame retardancy. The phenomenon of biomass composites prepared from the natural bamboo powder was the same, and there was still an obvious flame at 3 s, even though the densification treatment brought about by hot pressing had a flame-retardant effect in theory. However, due to the presence of hemicellulose, which confined lignin, and the poor interface compatibility between lignin and cellulose, the degree of densification was limited; therefore, the improvement effect on flame retardancy was not ideal. This poor flame retardancy was caused by the poor combination of materials.

After the alkali treatment, the flame retardancy of the material was improved to a certain extent, and the obvious flame did not appear until 6 s under the flame of the alcohol lamp, where it burned significantly after 9 s. The removal of hemicellulose made lignin a better precipitate on the surface of cellulose, and a better combination led to a denser structure, which made the combustion efficiency lower, thereby bringing a certain flame-retardant effect. The addition of L/C further improved the bonding of fibers, especially the 15% L/C sample, which still only had a small flame at 6 s, and it did not burn significantly at 9 s. It can be shown that a blend of sodium lignosulfonate and chitin, a by-product of papermaking, is also suitable as a flame retardant strategy for glue-free biomass composites [61].

### 2.4. Chemical Characterization Analysis of the L/C Bamboo Biocomposite

It can be seen from the comparison of the spectra of each sample in Figure 5a that the pretreated samples could see the disappearance of characteristic peaks at 1714 cm^−1^ and the weakening of the characteristic peak at 1239 cm^−1^, which corresponded to the weakening of the stretching vibration of carbonyl and C=O, which could be attributed to the removal of hemicellulose. The changes in 3313 cm^−1^ and 1024 cm^−1^ were attributed to the increase in -OH and C-H by the addition of chitosan. Changes caused by pretreatment could also be seen from the SEM (Figure 5d,e). After pretreatment, the surface became rough and shrunken, which reflected the removal of hemicellulose and cellulose amorphous regions. In the XPS test, it could be seen that the reduction in C-O, C=O and O-C-O, the increase in the proportion of C-C. and the increase in O/C also proved the removal of hemicellulose and the retention of the cellulose skeleton (Figure 5b,c). This removal brought about a loosening of the structure, allowing subsequent hot pressing to recombine into a tighter structure. Moreover, the loose structure also facilitated the introduction of the blend L/C, and the successful introduction of the blend was observed from SEM (Figure 5f). This successful introduction enhanced the stability of the material and reduced the waste of lignin waste liquid.

### 2.5. Waste Recycling and Environmental Analysis

Due to the non-adhesive strategy adopted by the prepared L/C bamboo composite and the introduction of L/C to improve the material properties, the material was made as a new type of environmentally friendly green biocomposite [62]. The formaldehyde emission of finished materials is one of the criteria for whether materials can be used in indoor scenes and whether they are harmful to human health. However, the formaldehyde emission of the L/C bamboo composite (0.01 mg/m^3^) is far lower than that of other commonly used wood-based panels, which is attributed to the material’s glue-free strategy (Figure 6a). The introduction of L/C blends to improve the mechanical properties and durability of biocomposites aimed at the disadvantages of and the low performance of glueless hot pressing made the performance meet the requirements of wood-based panels while having lower formaldehyde emission. Not only that but since the raw materials of the L/C bamboo composite were all recyclable natural biomass, there was no irreversible chemical change in the board preparation process; this enabled the complete recycling of the material, which was much higher than other wood-based panels using resin adhesives (Figure 6b). Due to the use of bamboo powder particles with a smaller particle size, the source of raw materials was entirely from waste bamboo and small bamboo branches, which greatly increased the source of raw materials.

The use of waste bamboo and sodium lignosulfonate reduced the possibility of these original wastes or industrial by-products harming the environment. Figure 6c shows the amount of waste consumed when producing 1 t of the L/C bamboo composite, the high-value utilization of 850 kg of waste bamboo and 830 kg of lignin black liquor (18–25% lignosulfonate content), which greatly alleviated environmental pollution and the waste of biomass resources. This made bamboo screening, pulp manufacturing, and L/C bamboo composite manufacturing a recyclable material utilization method, which met current society’s demand for green materials and the expectation of resource recycling. Therefore, the strategy of using sodium lignosulfonate and chitosan blends with waste bamboo to prepare biocomposites was green and environmentally friendly [63].

## 3. Materials and Methods

### 3.1. Source of Experimental Materials

Selected 60–100 mesh wood natural bamboo (*Phyllostachys heterocycle*) (China Guangdong Shenglong Bamboo & Wood Co., Ltd., Guangzhou, China) was used as the experimental raw material. Sodium hydroxide (≥96%) was used in the experiment and was purchased from Sinopharm Reagent Group Co., Ltd. (Shanghai, China) Chitosan, deacetylation degree ≥ 95%, was purchased from Shanghai Macklin Biochemical Technology Co., Ltd. (Shanghai, China) and sodium lignosulfonate was purchased from Shanghai Macklin Biochemical Technology Co., Ltd.

### 3.2. Pretreatment of the Bamboo Powder and the L/C Bamboo Biocomposite

The natural bamboo powder was dried in an oven at 60 °C; 8 g of bamboo powder was weighed, soaked in 100 mL of 10 wt% NaOH solution, and treated under hydrothermal conditions at 70 °C for 1 h. The product was subject to solid–liquid separation by vacuum filtration to obtain its powder, which was washed with deionized water until the powder was neutral.

Sodium lignosulfonate and chitosan were mixed into pretreated bamboo powder dried at 60 °C, and the ratio of sodium lignosulfonate to chitosan was 1:3. The blend was placed under hot pressing conditions of 186 °C and 110 MPa for 1 h to obtain L/C bamboo biocomposites. According to the additional amount of different sodium lignosulfonate/chitosan blends, the samples were named 5% L/C, 10% L/C, and 15% L/C, and the alkali-pretreated sample without the addition of the blend was named AKBC. The natural bamboo powder was treated with the same treatment method, and the prepared unbonded bamboo biocomposite (BC) was used as the control.

### 3.3. Measurements of Physical Properties of L/C Bamboo Biocomposite

The physical properties of biomass composites were tested by the requirements of GB/T 17657-2013 [64,65]. The mechanical strength (including static bending properties ((MOE and MOR)), tensile strength, and elastic modulus) of the material was analyzed by AGS-X universal testing machine from the Shimadzu Corporation of Japan.

The sample was placed on the balance to determine the mass, and then the volume of the absolute dry sample was determined by the vernier caliper, and the sample density was obtained by indirect calculation while the standard was referred to as GB/T 4472-2011 [66].

The sample was provided with a water environment to test the water resistance of the sample, and the weight and change in the sample was observed at 3, 6, 9, 12, 24, 24, and 48 h by immersing in water.

The water resistance of the biomass composite was verified by the contact–angle test. The DSA100S water contact angle instrument produced by the German KRUSS company was used for quantitative testing, and the appearance of the water droplet on the sample surface was saved by automatic photographing.

A scanning electron fiber microscope with an accelerating voltage of 15 kV was used to observe the microscopic morphology of the sample section.

DRPL-2B Thermal Constant Analyzer was used to test the thermal conductivity of the samples. The sensor calculated the thermal conductivity by detecting heat transfer through the sample from the hot electrode to the cooler. The sample size was limited to 20 × 20 × 2.5 mm.

The stability of samples under a flame was verified by direct burning with the outer flame of an alcohol lamp. A sample of 50 × 7 × 2.5 mm was placed in the flame for a certain period of time to observe the surface carbonization phenomenon.

The amount of formaldehyde released from the sample was tested according to GB/T 38794-2020 [44,67].

### 3.4. Functional Group Changes and Composition Changes of Biocomposites

The chemical functional groups of L/C bamboo biocomposites in the wavelength range of 4000–400 cm^−1^ by Nicolet IS50 Fourier transform infrared spectroscopy were combined with the attenuation total reflector (Thermo Fisher Scientific, Waltham, MA, USA).

The X-ray diffractometer manufactured by Beijing Puwei General Instrument Co., Ltd. (Beijing, China) was used to test the crystallinity of the sample, and an A current of 30 mA and a voltage of 40 kV were selected as operating parameters [68,69].

Axis Ultradld X-ray photoelectron spectroscopy (China Shimadzu Enterprise Management Co., Ltd., Beijing, China) was used to characterize the binding state and the relative amount of C and O atoms in the sample. The vacuum was set to 7 × 10^−8^ pa, and the voltage and current were selected as 15 kV and 10 mA, respectively [70,71].

## 4. Conclusions

In this paper, a special recycling strategy for papermaking waste liquid and waste bamboo is proposed. By using waste bamboo as raw material, a completely environmentally friendly biomass composite material was prepared by using glue-free hot molding. Aiming at the disadvantages of weak mechanical properties and the poor durability caused by the non-adhesive strategy, sodium lignosulfonate, a by-product extracted from papermaking black liquor, was used for modification. Sodium lignosulfonate and chitosan were blended into bamboo powder to modify the interfacial compatibility of lignin and cellulose, thereby promoting fiber bonding and improving material performance. In this way, papermaking black liquor and waste bamboo powder were converted into biomass composite materials. This high-value utilization method is effective. Each ton of biomass composite materials produced by this preparation method could reduce the waste of 850 kg of waste bamboo and the generation of 833 kg of papermaking waste liquid, effectively reducing environmental pollution. It meets the needs of the current society for the recycling of waste materials and environmental protection.

## Figures and Tables

**Figure 1 molecules-28-06058-f001:**
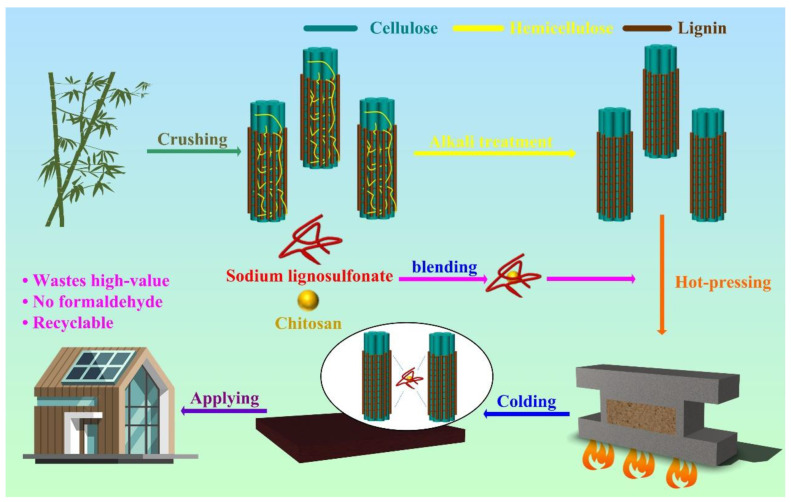
Preparation process of L/C bamboo biocomposite and changes in bamboo fiber.

**Figure 2 molecules-28-06058-f002:**
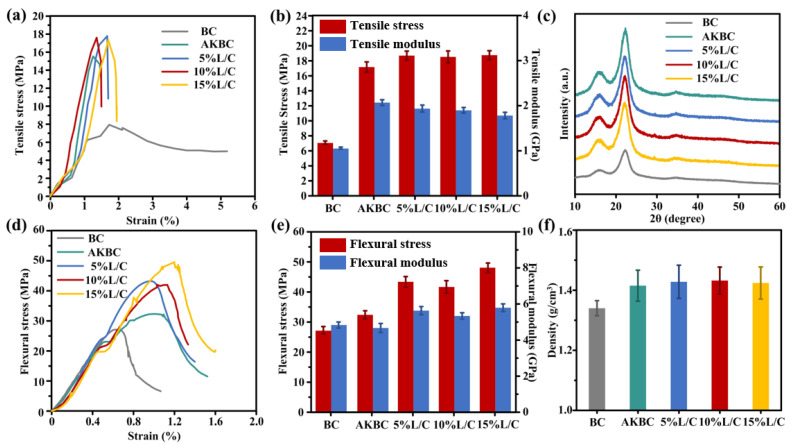
(**a**) Tensile strength-strain (**b**) Tensile strength and modulus, (**c**) XRD of the samples, (**d**) Flexural strength-strain, (**e**) Flexural strength and modulus, (**f**) Density.

**Figure 3 molecules-28-06058-f003:**
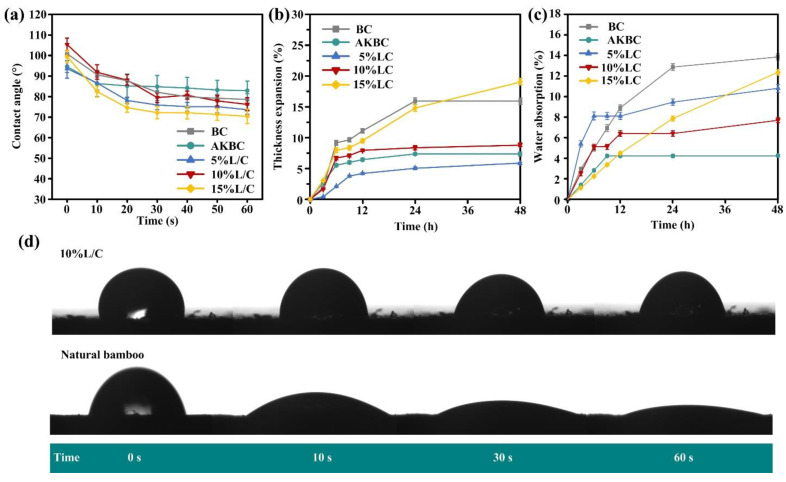
(**a**) The contact angle of the sample changes with time, (**b**) The contact angle of the sample changes with time, (**c**) The mass change in the sample in the water environment, (**d**) Water contact angle of natural bamboo and 10% L/C.

**Figure 4 molecules-28-06058-f004:**
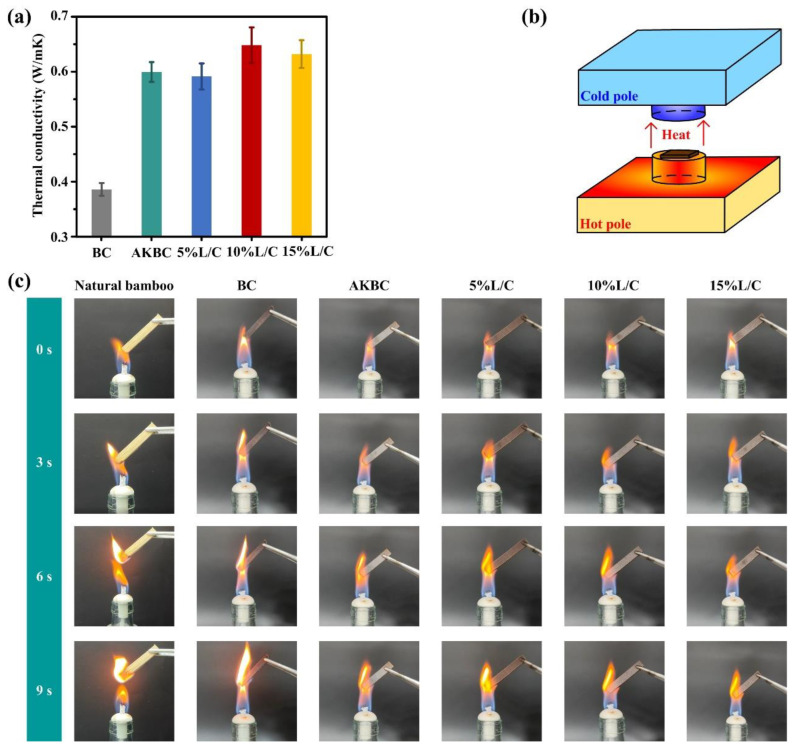
(**a**) Heat conduction of samples, (**b**) Schematic diagram of heat conduction test, (**c**) Combustion of each sample at 0 s, 3 s, 6 s, and 9 s.

**Figure 5 molecules-28-06058-f005:**
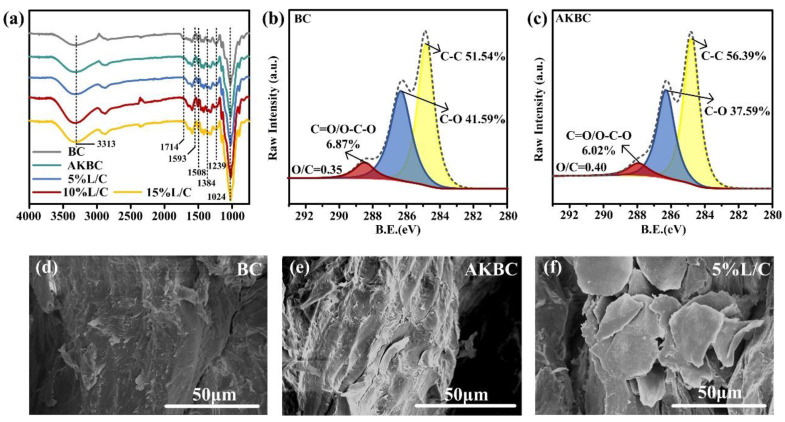
(**a**) FT-IR of the each sample, (**b**,**c**) XPS spectrum of the BC, AKBC, (**d**) SEM of the BC, (**e**) AKBC, (**f**) 5%L/C.

**Figure 6 molecules-28-06058-f006:**
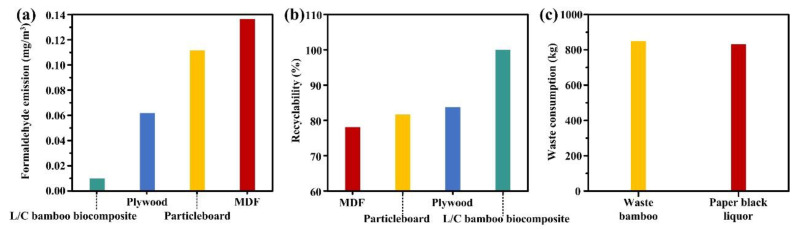
(**a**) Formaldehyde emission from L/C bamboo composite and wood-based panels, (**b**) Recyclability of L/C bamboo composite and wood-based panels, (**c**) 1 t of L/C bamboo composite prepared by the mass of waste consumed.

**Table 1 molecules-28-06058-t001:** Crystallinity of the biocomposite.

Sample	Crystallinity
BC	52.69%
AKBC	56.71%
5%L/C	54.14%
10%L/C	56.49%
15%L/C	58.29%

## Data Availability

The data presented in this study are available on request.

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
