# Peer review of "Recycling of Waste Bamboo Biomass and Papermaking Waste Liquid to Synthesize Sodium Lignosulfonate/Chitosan Glue-Free Biocomposite"

_molecules, 2023, doi:10.3390/molecules28166058_

Round 1
Reviewer 1 Report
Authors in this paper has presented strategy to synthesize novel glue free biocomposites with high-performance interfacial compatibility from papermaking waste products i.e. sodium lignosulfonate /chitosan and waste bamboo. Paper is well written and can be accepted for publication.
Author Response
Thank you very much for your recommend.

Reviewer 2 Report
The flame retardancy is suggested to be characterized by LOI and Vertical combustion method.
Minor editing is suggested.
Author Response
Comments and Suggestions for Authors:The flame retardancy is suggested to be characterized by LOI and Vertical combustion method.
Response: Thank you very much for your summary. Because the characterization of flame retardant ability is the verification of the binding effect of the material, it is only the evidence of the thermal conductivity and binding performance of the material. Therefore, we only chose the image of the direct burning of the flame of the alcohol lamp to directly reflect the effect, rather than selecting LOI and Vertical combustion method. We modified part of the statement to suit our needs. The modifications have been made on page 8, lines 253 in the manuscript in supporting information, as shown below:
The sample was placed under the flame of an alcohol lamp to observe its combustion. Similarly, the state of the sample under the flame of the alcohol lamp can also give back the same result (Fig .4c). As a comparison with natural bamboo, it is not difficult to see that there is already an obvious flame at 3 s, and it can burn independently from the flame of the alcohol lamp at 6 s and 9 s, which shows that natural biomass has poor flame retardancy. The phenomenon of biomass composites prepared from natural bamboo powder is the same, and there is still an obvious flame at 3s, even though the densification treatment brought about by hot pressing has a flame-retardant effect in theory. However, due to the presence of hemicellulose that confines lignin and the poor interface compatibility between lignin and cellulose, the degree of densification is limited, so the improvement effect on flame retardancy is not ideal. This poor flame retardancy is caused by the poor combination of materials.

Reviewer 3 Report
The main objective of this article is to develop adhesive-free biocomposites using bamboo/lignosulfonate as the raw material. The quality of the biocomposites produced was evaluated through a range of tests, including mechanical, physical, chemical, thermal, and formaldehyde release assessments. I recommend this manuscript for publication in the Molecules Journal. However, before accepting this manuscript for publishing, I suggest that the authors consider the following comments:
-Line 107 Please replace "bending strength" with "static bending properties (MOE and MOR)."
-Line 111 Please correct to "static bending strength and stiffness."
-Line 112 Please correct to "with a crosshead speed of..."
-Lines 126-128- Please add a reference for the method used to assess the stability of samples under flame.
- Line 133, Please indicate the purpose of using the X-ray diffractometer.
-Figure 2a. Please replace "tensile strength-strain" with "tensile stress-strain curve." The same applies to Figure 2d.
-Please include information on how the authors calculated density in the Materials and Methods section.
- What was the size of the bamboo powder used to produce the biocomposites?
- The authors did not provide any information on the methodology used to measure the formaldehyde emission, please, provide details on the testing process, including any relevant standards or protocols that were followed.
The article is written in good English, but a few typographical errors need to be corrected.
Author Response
Line 107 Please replace "bending strength" with "static bending properties (MOE and MOR)."
Response: We are grateful for the suggestion. We are sorry for the inconvenience caused by the unclear expression of the sentence. In order to better answer your question, we have added some sentences to make this data more convincing. Meanwhile, the modifications have been made on lines 107, as shown below:
The physical properties of biomass composites are tested by the requirements of GB/T 17657-2013. The mechanical strength (including static bending properties (MOE and MOR), tensile strength and elastic modulus) of the material was analyzed by AGS-X universal testing machine from Shimadzu Corporation of Japan.
Line 111 Please correct to "static bending strength and stiffness.
Response: We are grateful for the suggestion. We are sorry for the inconvenience caused by the unclear expression of the sentence. In order to better answer your question, we have added some sentences to make this data more convincing. Meanwhile, the modifications have been made on lines 111, as shown below:
The size of the sample in the tensile strength test is 50 × 7 × 2.2 mm, and the experiment is carried out with a crosshead speed of 0.2 cm/min and a span of 1.7 cm. A three-point bending test was used to test the static bending strength and stiffness with a crosshead speed of 1 cm/min and a span of 3 cm as parameters.
Line 112 Please correct to "with a crosshead speed of..."
Response: We are grateful for the suggestion. We are sorry for the inconvenience caused by the unclear expression of the sentence. In order to better answer your question, we have added some sentences to make this data more convincing. Meanwhile, the modifications have been made on lines 112, as shown below:
The size of the sample in the tensile strength test is 50 × 7 × 2.2 mm, and the experiment is carried out with a crosshead speed of 0.2 cm/min and a span of 1.7 cm. A three-point bending test was used to test the static bending strength and stiffness with a crosshead speed of 1 cm/min and a span of 3 cm as parameters.
Lines 126-128- Please add a reference for the method used to assess the stability of samples under flame.
Response: Thanks very much for the reviewer's comments, we have added the references in Line 126-128.
Stability of samples under flame was verified by direct burning with the outer flame of an alcohol lamp. A sample of 50 × 7 × 2.5 mm was placed in the flame for a certain period of time to observe the surface carbonization phenomenon.
Line 133, Please indicate the purpose of using the X-ray diffractometer.
Response: Thank you very much for the reviewer's request. We have added the purpose of using the X-ray diffractometer as required.
The X-ray diffractometer manufactured by Beijing Puwei General Instrument Co., Ltd. was used to test Crystallinity and crystallinity of the sample, and A current of 30 mA and a voltage of 40 kV were selected as operating parameters [42].
Figure 2a. Please replace "tensile strength-strain" with "tensile stress-strain curve." The same applies to Figure 2d.
Response: We have modified the manuscript according to the reviewer's requirements, and the modified part is in Figure 2a, 2d.
(a) Tensile strength-strain (b) Tensile strength and modulus, (c) XRD of the samples, (d) Flexural strength-strain, (e) Flexural strength and modulus, (f) Density.
Please include information on how the authors calculated density in the Materials and Methods section.
Response: We have modified the manuscript according to the reviewer's requirements, and the modified part is in Line 113.
The sample is placed on the balance to determine the mass, and then the volume of the absolute dry sample is determined by the vernier caliper, and the sample density is obtained by indirect calculation, the standard is referred to GB/T 4472-2011.
What was the size of the bamboo powder used to produce the biocomposites?
Response: Thank you very much for the reviewer's suggestion. We neglected the selection of raw materials in the manuscript, and we have revised it in line 86.
Selected 60-100 mesh wood natural bamboo (Phyllostachys heterocycle) (China Guangdong Shenglong Bamboo & Wood Co., Ltd.) was used as the experimental raw material.
The authors did not provide any information on the methodology used to measure the formaldehyde emission, please, provide details on the testing process, including any relevant standards or protocols that were followed.
Response: Thanks for the reviewer's suggestion. We are very sorry for such problems in the manuscript. We have provided formaldehyde test standards in the document, shown in Line 128.
The amount of formaldehyde released from the sample was tested according to GB/T 38794-2020 [44].
- Ge, S.; Ouyang, H.; Ye, H.; Shi, Y.; Sheng, Y.; Peng, W., High-performance and environmentally friendly acrylonitrile butadiene styrene/wood composite for versatile applications in furniture and construction. Advanced Composites and Hybrid Materials 2023,6, (1).

Reviewer 4 Report
only some minor changes
as mentioned, the one statement 850+830 =1000 can not be true and needs to be changed / explained
and also the question is really, what is the benefit of adding >5%

see attached file
Author Response
only some minor changes, as mentioned, the one statement 850+830 =1000 can not be true and needs to be changed / explained, and also the question is really, what is the benefit of adding >5%.
Response: Thank you very much for the comments of the reviewers. In the section of calculating waste consumption, 830kg lignin black liquor refers to the original black liquor produced by the paper mill, which contains a variety of substances and is generally considered to contain 18 - 25% lignosulfonate content. We have made changes in the document to better highlight our meaning. More control groups could show a change in performance due to the addition of the blend, so we chose a variety of experimental groups for comparison.
Fig 6c shows the amount of waste consumed when producing 1 t of L/C bamboo composite, the high-value utilization of 850 kg of waste bamboo and 830 kg of lignin black liquor (18%-25% lignosulfonate content) has greatly alleviated environmental pollution and waste of biomass resources.
I need three attempts of reading that all samples except BC are AKBC, maybe the language should be clearer
Response: Thank you very much for the reviewer's comments. We have modified the name of the sample in the manuscript to make it clearer.
The blend was placed under hot pressing conditions of 186 °C and 110 MPa for 1 h to obtain L/C bamboo biocomposites. According to the addition amount of different sodium lignosulfonate/chitosan blends, the samples were named, 5% L/C, 10% L/C, 15% L/C, the alkali-pretreated sample without the addition of the blend is named AKBC. The natural bamboo powder was treated with the same treatment method, and the prepared unbonded bamboo biocomposite (BC) was used as the control.
I assume d is flexural stress
Response: Thank you very much for the reviewer's comments. We are very sorry for such problems in the manuscript, and we have revised the manuscript.
I agree L/C blends are better, but the amount 5-15% does not really make a different.
Response: Thank you very much for the reviewer's suggestions, but there are still some subtle differences in the different proportions of blends even if the gap is not large, this can be shown in subsequent performance tests.
No red in the legend, is it also 10%?
Response: Thanks very much for the reviewer's comments, we have modified the picture.
This statement seems to ignore the law of constant masses 850 + 830 = 1000? Where is the rest??
Response: Thank you very much for the comments of the reviewers. In the section of calculating waste consumption, 830kg lignin black liquor refers to the original black liquor produced by the paper mill, which contains a variety of substances and is generally considered to contain 18 - 25% lignosulfonate content. We have made changes in the document to better highlight our meaning.
Fig 6c shows the amount of waste consumed when producing 1 t of L/C bamboo composite, the high-value utilization of 850 kg of waste bamboo and 830 kg of lignin black liquor (18%-25% lignosulfonate content) has greatly alleviated environmental pollution and waste of biomass resources.

Round 2
Reviewer 2 Report
The author addressed my concerns.
Minor editing is suggested.